# The Impact of Telepresence Robots on Family Caregivers and Residents in Long-Term Care

**DOI:** 10.3390/ijerph22050713

**Published:** 2025-05-01

**Authors:** Lillian Hung, Joey Oi Yee Wong, Haopu (Lily) Ren, Yong Zhao, Jason Jincheng Fu, Jim Mann, Lun Li

**Affiliations:** 1IDEA Lab, School of Nursing, The University of British Columbia, Vancouver, BC V6T 2B5, Canada; 2School of Nursing, The University of British Columbia, Vancouver, BC V6T 2B5, Canada; 3School of Interdisciplinary Studies, The University of British Columbia, Vancouver, BC V6T 2B5, Canada; 4School of Biomedical Engineering, The University of British Columbia, Vancouver, BC V6T 2B5, Canada; 5Department of Social Work, MacEwan University, Edmonton, AB T5H 0K9, Canada

**Keywords:** telepresence robots, long-term care, quality of life, loneliness, caregiver burden

## Abstract

Telepresence robots can enhance social connection and support person-centered care in long-term care (LTC) homes. This study evaluates their impact in facilitating virtual visits between family caregivers and older residents in Canadian LTC homes. Telepresence robots were placed in residents’ rooms, allowing virtual visits at mutual convenience. A total of 18 residents and 17 family caregivers participated. Quantitative assessments included the Zarit Burden Interview, the De Jong Gierveld Loneliness Scale, and the Quality of Life in Alzheimer’s Disease scale, while qualitative data were collected through interviews, field notes, and observations. Repeated ANOVA showed that using telepresence robots significantly reduced caregiver burden (*p* = 0.008), improved residents’ quality of life (*p* = 0.028), and decreased resident loneliness (*p* = 0.038). Older caregivers experienced the greatest burden reduction, with scores dropping from a mean of 25.0 at baseline to 16.1 at two months. Qualitative findings provided further context, revealing that residents felt more connected, close, and engaged, while families found the robots facilitated continuity of care, complemented in-person visits, reduced stress and guilt, and provided reassurance. These findings suggest that telepresence robots can enhance the well-being of both residents and caregivers in LTC homes, though future research should explore their long-term impact and technological limitations.

## 1. Introduction

Loneliness in long-term care (LTC) remains an enduring and multifaceted challenge, intensified by the COVID-19 pandemic [1]. For residents in LTC settings, loneliness may extend beyond mere physical isolation, encompassing social disconnection and feelings of abandonment [2,3]. These factors collectively contribute to adverse health outcomes, including cognitive decline, depression, and reduced quality of life. Emerging evidence suggests that older men may be particularly vulnerable to loneliness in LTC settings. For example, Dahlberg et al. [4] and Wright et al. [5] described that older men, especially those who have lost social supports such as spouses, report higher levels of loneliness compared to women. This heightened vulnerability can be attributed to a confluence of factors, including the erosion of traditional social roles following retirement, a comparatively narrower social network, and societal expectations that discourage emotional expressiveness among men [6,7]. These issues may be further exacerbated by limited opportunities for spontaneous social engagement and meaningful interactions.

Family caregivers often experience substantial burdens during the transition of loved ones into LTC homes [8]. The reconfiguration of caregiving roles, from primary caregiver to a more peripheral yet emotionally invested visitor, faces unique challenges, for example, the feelings of guilt about placing their loved one into LTC, the loss of lifelong company at home, and the disrupted sense of purpose as a caregiver [9,10]. Such a caregiver burden not only undermines the well-being of family members but also impacts residents’ quality of life. Strong family relationships are associated with enhanced emotional well-being, greater social connectedness, and improved overall quality of life among LTC residents [11].

In response to these dual challenges—resident loneliness and caregiver burden—technology-driven interventions offer promising solutions to facilitate meaningful social connections. In this context, we have explored a more accessible option for virtual video calls by using telepresence robots [12]. These robotic systems allow for immersive, remote-controlled virtual visits, enabling easy, personalized interactions. The telepresence robot adopted in our study consists of a tablet (the head), a pole (the body), and wheels (the base). The tablet consists of a screen, a camera, and speakers for conversations; the pole allows adjustments of the robot’s height to facilitate communication; the wheels allow the robot to move in all directions (see Figure 1). The telepresence robot is fully controlled and driven by the callers, i.e., family members. With a device with a camera and microphone (e.g., smartphones, tablets, and laptops) and an Internet connection, the family members can call in via the telepresence robot. The call recipients (i.e., residents) do not need to press any buttons to receive the calls. However, there is an “end call” function on the screen for them to discontinue the call if necessary. Although all allow real-time communications, telepresence robots in LTC have benefits compared to traditional approaches, such as telephone calls or video conferencing via handheld devices. These traditional approaches often fall short due to technological barriers, sensory impairments, and the reliance on staff for operational assistance [13,14]. As family members can drive the telepresence robots toward the resident and start the calls, residents are not required to recognize the ringtone, look for or hold a device, or press the right button to pick up calls [15]. Unlike regular phone or video calls, these hand-free telepresence robots do not require staff to be present and assist during the calls. Family members can interact more and even perform exercises with residents. Without staff presence, privacy can be better maintained during virtual calls. Family members can call in at a more flexible frequency and for a longer time [13,14].

Recent advancements in telehealth and AI-driven assistive technologies have introduced diverse tools to support the social and emotional well-being of older adults in LTC environments. Innovations such as socially assistive robots, AI chatbots, and remote monitoring systems have shown promise in alleviating loneliness, enhancing caregiver engagement, and improving quality of life. For instance, socially assistive robots like PARO and LOVOT have demonstrated measurable reductions in loneliness and depression among LTC residents by promoting meaningful engagement and companionship [16,17]. Similarly, AI-powered chatbots and voice assistants are being piloted to provide older adults with on-demand emotional support and routine social interaction, with early evidence suggesting improvements in perceived connectedness and mood [18]. A recent systematic review also highlighted the role of smart home and remote monitoring technologies in reducing caregiver burden and fostering more timely, person-centered responses in LTC and dementia care [19]. Even though the technological design of telepresence robots may no longer be novel, their application in real-world LTC settings using a longitudinal mixed-methods approach remains limited.

Previous studies have demonstrated that telepresence robots help maintain social bonds, enhance residents’ sense of being valued and remembered, and support person-centered care [15,20]. From the residents’ perspective, telepresence robots nurture emotional closeness and meaningful engagement, while family members experience reduced stress, alleviated feelings of guilt, and reassurance regarding their loved one’s well-being.

While emerging qualitative evidence highlights the positive impact of telepresence robots, several concerns and barriers to implementation have also been noted [21,22]. Care facility staff have expressed concerns regarding potential resident confusion, privacy issues, and their own lack of skills and additional workload associated with managing the robots. Thus, systematic longitudinal assessments are needed to understand the long-term implications of using telepresence robots in LTC settings. This study addresses that gap by systematically examining the impact of telepresence robots on loneliness and quality of life among LTC residents, as well as the caregiver burden of their family caregivers. By integrating quantitative measures with rich qualitative insights, we aim to understand the broader implications of technology-driven interventions for improving the emotional and social well-being of older adults in LTC and their family caregivers. Our research questions are the following:What are the experiences of residents and families in using telepresence robots?Does the telepresence robot reduce loneliness, improve the quality of life of residents in LTC homes, and reduce the burden for family caregivers?

## 2. Methods

### 2.1. Design

We adopted a mixed-methods design, a convergent parallel mixed-methods approach under which quantitative and qualitative data were collected simultaneously and analyzed separately and independently [23] (p. 280) [24] (p. 519). Quantitative methods are advantageous in showing what relationships are established among variables; in contrast, qualitative methods are powerful in showing how and why these relationships develop [24] (p. 519). With quantitative and qualitative data complementing each other, the approach allows us to draw on the strengths of both methods for a more comprehensive understanding of our research focus, which is the impact of using telepresence robots on the emotional and social well-being of older adults in LTC and their family caregivers [25] (p. 318). The approach also enables us to compare findings from both datasets to gain better insights into the breadth and depth of individual experiences with telepresence robots and contextual factors [26]. Figure 2 shows the convergent parallel mixed-methods approach used in this study [27] (p. 218). Refer to Appendix A for the COREQ checklist and Appendix A for the GRAMMS checklist.

### 2.2. Data Collection

#### 2.2.1. Research Instruments

We used three survey instruments for the quantitative study: Zarit Burden Interview, DeJong Gierveld Loneliness Scale, and Quality of Life—Alzheimer’s Disease (QOL-AD) scale. Firstly, the Zarit Burden Interview (12 items) [28] was used to measure caregiver burdens. The questionnaire has questions such as “How often do you feel that because of the time you spend with your relative, you do not have enough time for yourself?” and “How often do you feel angry when you are around your relative?”. Participants could choose from “never”, “rarely”, “sometimes”, “quite frequently”, or “nearly always” (scored from 0 to 4, respectively) to answer. The higher the total score, the greater the burden on the family caregiver. Secondly, the 6-item DeJong Gierveld Loneliness Scale [29] was used to measure loneliness, with 3 items corresponding to emotional loneliness and social loneliness each. Participants could choose from “Yes”, “More or less”, and “No” for an answer that best described their feelings. Items for emotional loneliness were negatively worded (e.g., I experience a general sense of emptiness), and answers of “Yes” or “More or less” were scored three or two and “No” scored one. Questions for social loneliness were positively worded (e.g., There are many people I can trust completely), and the scoring was reversed. A higher score represents greater loneliness. Lastly, the QOL-AD scale was used to measure participants’ quality of life [30]. The QOL-AD contains 13 items related to a group of life domains, including physical health, relationships, activities, and capabilities to perform some daily tasks. Participants could answer each question by choosing an answer from “poor”, “fair”, and “good” to “excellent”, with “poor” assigned 1 point and excellent assigned 4 points. A higher total score means a higher quality of life.

For qualitative data collection, we conducted semi-structured interviews with our participants as we asked them questions on the abovementioned scales. With questions such as “Can you tell me more about your experience of using the robot?” and “Can you share some stories with us?” semi-structured interviews allowed us to follow up on experiences raised by our participants and gain rich, in-depth information about our study focus [31] (p. 36). We also observed them during the data collection and took field notes to document our real-time observations and reflections, adding contextual depth to our data [32]. These qualitative methods complemented quantitative data to enrich the evidence on our research topic and enhanced the overall quality of our quantitative and qualitative data collected [33].

#### 2.2.2. Setting

This study is part of a larger, three-year project that explores the experiences of residents, family caregivers, and staff in implementing telepresence robots in LTC settings for older people with dementia. We placed 20 robots in five LTC homes in an urban area of British Columbia, Canada. See Table 1 for detailed information about our partnered care sites.

#### 2.2.3. Participant Recruitment and Sampling

We recruited participants with purposive sampling. The research team collaborated with staff champions from the recreation and rehabilitation departments of each care site to introduce the telepresence robot to residents and family caregivers and explain the study. Participants were intentionally recruited to reflect diverse racial and ethnocultural backgrounds, genders (male and female), and various age groups. We purposefully included residents with different mobility impairments requiring assistive devices like wheelchairs or walkers, as well as those living with various forms and stages of dementia. We had no participant drop-off during the two-month intervention.

This pre–post design, utilizing each participant as their own control, enhances statistical power even with a smaller sample size. The Partial Eta-Squared (η^2^p) values for caregiver burden (0.318), quality of life (0.191), and loneliness (0.203) in this study show large effect sizes, indicating substantial within-subject changes due to the intervention. Using G*Power (version 3.1) with set parameters (α = 0.05, 1 − β = 0.80, and 3 measurements), we determined that the recommended sample size was 9, based on Cohen’s f calculated from the Partial Eta-Squared (η^2^p) values provided above. Given these strong effect sizes, the sample size of 35 (18 residents and 17 family caregivers) provided adequate power to detect meaningful changes.

#### 2.2.4. Timeline

Data collection for this paper was performed between December 2022 and September 2023, which was the third phase of the larger project. Data collection for each participant lasted eight weeks (see Figure 3). For residents, we collected data on their quality of life and loneliness at three points in time: before they started using the robot (T_0_), four weeks after using the robot (T_1_), and eight weeks after using the robot (T_2_). Similarly, we collected data on the caregiving burden from family caregivers at T_0_, T_1_, and T_2_. Notably, every participant’s T_0_ starts at a different time, as they started using the robot at a different time.

We interviewed all residents in person as they preferred this approach. We scheduled data collection at a time and location at their convenience, often when they felt most energetic in the day and in their individual room at the care site. For family caregivers, we interviewed some of them in person during their visits to their loved ones in the LTC home. Interviews with other family caregivers were conducted remotely over Zoom or phone. Each interview lasted between 20 and 60 min. Data collected were recorded and transcribed verbatim. Data collection ceased when the research team agreed that the data were sufficient to answer our research questions.

#### 2.2.5. The Research Team

Our team consists of multidisciplinary researchers and trainees. The Principal Investigator (PI), LH, is a professor in the School of Nursing at the University of British Columbia. LL is a professor in the School of Social Work at MacEwan University. The PI oversees the project and trains undergraduate and graduate trainees. JW, YZ, and LR are PhD students in nursing, medical science, and occupational therapy. JF is an undergraduate student in biomedical engineering. The team also involves patient partners living with dementia and family caregivers who contribute lived experience expertise. Weekly research meetings allowed the team to discuss findings, critically challenge assumptions, and deepen our understanding of the research topic from different perspectives. JW, LR, and JF collected quantitative and qualitative data simultaneously. LL and YZ performed quantitative data analysis, while JW and LR led qualitative analysis.

### 2.3. Data Analysis

#### 2.3.1. Quantitative Data Analysis

Statistical analyses were conducted using SPSS (version 29.0.2.0), with a significance level set at 0.05 for all two-tailed tests. First, descriptive analyses were performed to examine gender, age group, ethnicity, and family relationships, reporting counts and percentages to provide a comprehensive view of participants’ demographic characteristics. Descriptive statistics were then used to calculate the mean and standard deviation for the three scales (caregiver burden, quality of life, and loneliness) across three time points.

A repeated-measures analysis of variance (ANOVA) was conducted to assess changes in each scale’s scores over time within the entire sample. If the assumption of sphericity was violated, a Greenhouse–Geisser correction was applied. When a significant effect was detected, Bonferroni post hoc tests were performed to determine which time points significantly differed. This analysis was then repeated for key sociodemographic subgroups, including binary gender (male and female), ethnicity (Asian and white), and age groups (for the caregiver burden scale only: younger than 60 years vs. 60 years and older). Cronbach’s alpha values indicated acceptable to high internal consistency for the scales. For the caregiver burden scale, reliability was 0.932 at Time 1, 0.895 at Time 2, and 0.870 at Time 3. For the loneliness scale, Cronbach’s alpha was 0.745 at Time 1, 0.643 at Time 2, and 0.654 at Time 3. The quality-of-life scale showed reliability values of 0.853 at Time 1, 0.804 at Time 2, and 0.911 at Time 3.

#### 2.3.2. Qualitative Data Analysis

We followed the reflexive thematic analysis process proposed by Braun and Clark [34,35]. Step 1: LH trained the research team in mixed data analysis. Step 2: JW and LR familiarized themselves with the qualitative data. Step 3: JW and LR coded the qualitative data manually and developed initial themes (see Table 2). Step 4: In the research meeting, JW and LR presented themes, and the research team discussed and reviewed the themes. Step 5: Informed by the team discussion, the two trainees, JW and LR, refined the themes. Step 6: The team approved the updated themes and agreed on the final draft of the manuscript.

### 2.4. Data Integration

We compared, merged, and interpreted quantitative and qualitative data to identify points of alignment and differences in findings [36]. Then, we developed narratives to explain how our quantitative and qualitative data complemented each other in the discussion, providing richer context and deeper insight.

### 2.5. Rigour

In our regular research meetings, we consistently involved patient partners, family caregivers, and clinicians to remain reflexive about our positions and assumptions about our research activities and participants. We involved patient partners in the pilot testing survey and interview questions and reframed them accordingly to make these questions understandable for participants, especially residents with dementia. We adopted multiple data collection methods, including surveys, interviews, field notes, and observations, to gain a more comprehensive understanding of the research topic. In addition, LH trained the trainees using different methods to collect data from participants respectfully and so the data collection was conversational and accessible.

### 2.6. Ethics

This study was approved by the Behavioral Research Ethics Board of the University of British Columbia on 28 September 2022. All our participants gave the research team informed consent before participating in this study. Pseudonyms were used for participants and LTC homes to protect confidentiality.

## 3. Results

This study included 35 participants, comprising 18 residents and 17 family members (see Table 3). Among the residents, males accounted for 55.6%, while females made up 44.4%. The majority of the residents were aged 81 years or older (66.7%), followed by those aged 71–80 years (27.8%) and a smaller proportion aged 61–70 years (5.6%). In terms of ethnicity, 61.1% of residents identified as Asian, while 38.9% were white. 

Among the family members, the gender distribution was predominantly female (82.3%), with males making up 17.6%. Family members’ ages varied, with the largest proportion in the 61–70 age group (29.4%), followed by 51–60 years (23.5%) and 31–50 years (23.5%). Smaller proportions were in the 71–80 (11.8%) and 21–40 (11.8%) age groups. Ethnically, 58.8% of family members identified as Asian, while 41.2% were white. Most of the caregivers were daughters to residents (64.7%), followed by sons and other roles (spouse, granddaughter, friend).

### 3.1. Quantitative Results

The results in Table 4 show significant changes in family caregiver burden, residents’ quality of life, and loneliness over three time points: baseline (T_0_), 1 month after using a robot (T_1_), and 2 months after using a robot (T_2_). For family caregiver burden, the mean decreased from 21.5 ± 11.09 at T_0_ to 16.5 ± 7.92 at T_1_ and further to 15.8 ± 7.68 at T_2_. ANOVA revealed a significant overall difference (F(2) = 7.464, *p* = 0.008, partial η^2^ = 0.318), indicating a large effect size. Post hoc comparisons showed significant reductions between T_0_ and T_1_ (*p* = 0.011) and T_0_ and T_2_ (*p* = 0.043). Subgroup analysis (Figure 4) revealed no statistical differences at T_0_ between subgroups based on gender, age, or ethnicity, although differences in mean values were observed. The highest burden score of 40 was reported by a spouse, followed by daughters with a mean of 22.1 and sons with a mean of 23.7, compared to a granddaughter with 8 and a friend with 2. Male caregiver burden was also notably higher at 23.7 than in female caregivers at 21.0 (Figure 4). Female caregivers (*n* = 14) showed similar levels and trends to the total group, with a baseline score of 21.0, which decreased to 16.0 at 1 month and 15.2 at 2 months (*p* = 0.025). Caregivers over 60 reported a higher caregiver burden at all three time points compared to those under 60, although the differences were not statistically significant. The older adult subgroup began with the highest burden score of 25.0 at baseline, which decreased to 18.9 at 1 month and 16.1 at 2 months (*p* = 0.01). These findings suggest a significant reduction in caregiver burden over time across all groups, with the greatest benefit observed in the older adult caregiver subgroup.

Regarding quality of life, the mean increased from 29.9 ± 6.93 at T_0_ to 31.6 ± 6.55 at T_1_ and 32.2 ± 7.62 at T_2_. Although ANOVA showed a significant overall difference (F(2) = 4.024, *p* = 0.027, partial η^2^ = 0.191), post hoc comparisons did not reveal significant differences between any pair of time points (Table 4). Subgroup analysis (Figure 5) indicated similar trends across different demographic groups without statistically significant differences, regardless of gender or ethnicity. Male residents and white residents consistently reported higher quality of life scores than female and Asian counterparts without statistically significant differences.

For resident loneliness scores, the mean decreased from 11.1 ± 3.43 at T_0_ to 9.9 ± 2.99 at T_1_ and slightly to 9.8 ± 2.73 at T_2_ (F(2) = 4.084, *p* = 0.026, partial η^2^ = 0.203), with no further significant pairwise differences found between any time points (Table 4). Subgroup analysis (Figure 6) revealed a similar pattern across all groups. Specifically, within the ethnic subgroups, Asian residents experienced a greater decrease in loneliness, with scores dropping from 12.5 at T_0_ to 10.7 at T_1_, followed by a slight increase to 11.0 at T_2_ (*p* = 0.030). In contrast, the white subgroup did not report a reduction in loneliness at T_1_ but showed a significant decrease at T_2_, which differed from the trend observed in other subgroups. No significant differences in loneliness scores were found between male and female subgroups. These findings suggest that while loneliness decreased over time for all groups, the changes varied by ethnicity, with white residents showing a delayed improvement in loneliness.

Overall, a reduction in caregiver burden and an improvement in quality of life and loneliness were reported during the implementation of the telepresence robot. Most of these improvements occurred within the first month after initiation and were sustained through the second month.

### 3.2. Qualitative Results

The analysis of the qualitative data generated three themes on the impact of telepresence robots on residents and family caregivers (refer to Table 5 for a summary of themes).

#### 3.2.1. Impact on Residents

##### Promotes Connections and Nurtures Closeness

With their function for video calls, telepresence robots connect residents with their families and friends, spanning multiple generations. Family caregivers described sharing everyday experiences such as artwork, music, and family events. One family caregiver who lived far from the LTC home shared how she showed her artwork and played ukulele for her dad through the telepresence robot. She said, “*He can get a feel of what is going on in my life*” (Candy, daughter of resident, white, Sunflower Residence). Residents expressed joy and closeness when engaging with the robot. A resident shared, “*I have four grandchildren. Two of them will call in through the robot… Usually, we will talk for a long time*” (Winky, a resident, Asian, Tulip Residence). A family caregiver shared family moments through the robot with her father. She expressed, “*When I show him the Christmas tree and different things going on in my life, he is happy. One time my sister showed my dad her daughter’s ballet dance, he was so happy*” (Cecilia, daughter of a resident, Asian, Tulip Residence).

Residents also preferred visual interactions offered by robots compared to traditional phone calls. For example, a resident stated, “*The robot allows me to see the face of my daughter. Yes, I feel somehow happy seeing my daughter. There are differences between the phone and the robot because I can see her*” (Cherry, a resident, Asian, Tulip Residence).

##### Provides Meaningful Engagement

Telepresence robots offered residents opportunities for meaningful interactions and conversations, significantly reducing feelings of isolation. A resident expressed, “*There are no people that I can chat with [in the LTC home]. On the robot, my granddaughter will ask me where I feel pain. I feel happy to talk to her through the robot*” (Winky, a resident, Asian, Tulip Residence). A daughter shared, “*With more frequent calls through the robot, my dad can recall our conversations. He can chat and talk forever. It feels like an in-person visit to him*” (Flora, daughter of a resident, white, Rose Residence). One caregiver described how frequent robot interactions enhanced a resident’s “sense of self and self-worth”. Another emphasized how visual interaction helped adapt communication strategies effectively. A family member shared, “*The robot has helped in such a way that I can see her [the resident] when I’m asking her a question. I can see if she’s struggling with the answer, and then that means that I need to rephrase the question or talk about something different*” (Shereen, daughter of a resident, white, Sunflower Residence).

##### Acknowledges Robots’ Limitations

A few family members voiced the limitations of telepresence robots. As the family caregivers are the main controllers of the telepresence robots, the residents cannot initiate calls. The residents’ role is more passive in the call via telepresence robots. Therefore, some residents recognized the lack of control over the robot. A resident remarked, “*If I don’t have the robot, that’s no big deal to me because I didn’t have this robot in my whole life. I can use the phone to call my family*” (Paul, a resident, Asian, Tulip Residence). Another resident commented, “*The robot will move towards me when I am sitting. I don’t know how to use the robot. I feel annoyed with myself. I feel helpless. I feel useless. I have so much pain in my legs. No one can help me, not even the robot*” (Cherry, a resident, Asian, Tulip Residence). There were also periods when the telepresence robots had technical issues (e.g., mobility issues, inability to adjust the height) and needed to be sent for repairs. A family member shared an experience: “*When Kirby [name of the telepresence robot] had a tech issue, we [the resident and herself] joked about Kirby having a glitch and needing some downtime or vitamins. We had a good laugh*” (Lily, wife of a resident, white, Jasmine Residence).

#### 3.2.2. Impact on Family Caregivers

##### Reduces Stress and Guilt

Family caregivers reported that the robot reduced stress and guilt associated with challenges in visiting LTC homes, especially during adverse weather and personal illness. Telepresence robots provided a practical alternative, easing caregiving burdens. One caregiver shared relief over less commuting, stating, “*With the snow, I am not able to see her [the resident] for a while. I think about her sitting in the nursing home by herself*” (Catherine, friend of a resident, white, Jasmine Residence). Another family caregiver said, “*I felt bad not seeing him [the resident] every day as he was expecting me. After I had been sick, it was becoming more challenging to maintain my visits*” (Lily, wife of a resident, white, Jasmine Residence). A resident’s son said, “*The key benefit of the robot is time-saving for commute. The robot helps me to show up and be there more with my father*” (Jeffrey, son of a resident, Asian, Tulip Residence). Another described, “*The robot helps with relieving the pressure of driving. I have two young kids, and it’s hard having the time to drive to him [the resident] with two hours of travel*” (Janice, daughter of a resident, white, Zinnia Residence). Another family caregiver also shared how the telepresence robot supported her “visits to her mum” when she was sick. She said, “*I was sick three weeks ago. I had a fever. Thank goodness I have the robot. I can visit my mom*” (Ruby, daughter of a resident, Asian, Tulip Residence).

##### Enables Continuity in Caregiver’s Role and Complements In-Person Visits

Transferring to LTC homes can be a challenging transition period for residents and family caregivers. Many caregivers in our study described how telepresence robots support the continuation of their caregiving roles and their involvement in the care of residents in LTC homes.

A daughter of a resident illustrated this continuity: “*The robot helps with the feeling that I am still looking after Mom, seeing what she’s doing, and making sure that she knows that I am around. I can go on the robot and see her. And I know her routine. At least to say, ‘Hello mum’*” (Ruby, daughter of a resident, Asian, Tulip Residence).

A resident’s son also shared how he supported staff in the resident’s care. He said, “Before, when the staff was helping my dad, he would get very scared because he didn’t know who they were. With the telepresence robot, I can call in and tell him who they [the staff] are, and he feels better. He knows that his family is around, and calms down” (Jeffrey, son of a resident, Asian, Tulip Residence).

Another caregiver shared how telepresence robots allowed advocacy for their loved ones in LTC during the COVID-19 pandemic: “*The robot was really good when she [the resident] was confined because of the COVID outbreak… She was confused and distressed. After I got off [logged out] from the robot, I asked the staff to talk to her. It was helpful as I could not go into the nursing home*” (Catherine, friend of a resident, white, Jasmine Residence).

Notably, most family caregivers emphasized that despite the presence of telepresence robots, they were still eager to visit their loved ones and have a personal touch. One caregiver said, “*Even though I use the robot, I still come three times a week to visit the care home. The robot doesn’t replace that time*” (Ruby, daughter of a resident, Asian, Tulip Residence). Another caregiver echoed the preference of his mother, “*The robot is a great addition. [But] for my mom, she would prefer an in-person visit*” (Louis, son of a resident, Asian, Rose Residence).

##### Offers Reassurance on Residents’ Quality of Life

The telepresence robots helped reassure family caregivers about enhancing residents’ quality of life in LTC. One family caregiver shared, “*The staff told me that [the resident] is calmer and not as lonely, more settled*” (Lily, wife of a resident, white, Jasmine Residence). Another family caregiver described, “*The robot really helps with his spirits… It’s obviously better*” (Nancy, daughter of a resident, Asian, Jasmine Residence).

Knowing the benefits of the telepresence robots on residents had a positive impact on the caregivers. One of them smiled and said, “*I think it helps mum… She is quite happy when she sees me, even though I am on the robot. Knowing that she is at ease is good for me too. I feel happier knowing that Mum is happy seeing me. I can tell that she is happy from her expression. She is not very verbal. For mum, she will be more grounded if she sees me more*” (Ruby, daughter of a resident, Asian, Tulip Residence).

## 4. Discussion and Implications

This study examined the impact of a telepresence robot intervention on the family caregiver burden, residents’ quality of life, and loneliness. The findings demonstrated a significant reduction in caregiver burden, particularly during the first month of implementation, indicating immediate relief through improved communication, reduced logistical challenges, and increased reassurance for caregivers. Our qualitative insights reinforced the quantitative results, revealing reduced caregiver stress and guilt as caregivers provided emotional support and reduced residents’ anxiety using the telepresence robots. These findings align with Marvardi et al. [37], who highlighted the caregiver burden as primarily influenced by residents’ behavioral disturbances and disability, alongside caregiver anxiety and depression [37]. Their findings align with our qualitative themes, which highlight the reduction in caregiver burden when caregivers can help decrease residents’ anxiety, calm residents down, and provide emotional support for residents by calling in with the telepresence robot. Importantly, while caregivers in our study emphasized that telepresence robots would not replace in-person visits, these robots offered additional benefits, such as reducing the travel time, which reduced caregivers’ stress and guilt.

Previous research identified the caregiver burden as closely related to residents’ neuropsychiatric symptoms and caregivers’ competence, with emotional burdens exacerbated by residents’ high affiliation and caregiver gender [38]. Our findings partially supported these conclusions. The highest burden scores in our study were reported by one individual spouse and older caregivers’ subgroup, aligning with earlier findings by Rinaldi et al. [39], in which older caregivers experienced stress levels three times higher than younger groups, and children or spouses reported burden-distress five to six times higher than other relatives. Although our study did not find statistically significant differences based on residents’ gender or caregiver affiliation due to the limited sample size, the highest burden score was reported by a spouse, followed by daughters and sons, compared to a granddaughter and a friend, which aligns with the above two research studies [38,39]. Notably, male caregivers reported higher initial burden levels than female caregivers, which is also consistent with prior research [39]. Older caregivers, who initially reported the highest burden, demonstrated the greatest reduction in burden following the intervention. This trend suggests that older caregivers, often facing higher stress and physical strain, may benefit most from interventions that ease perceived caregiving demands. An older adult spouse in our study expressed that hospitalization was once a barrier for her to visit the LTC home, which led to the feeling of guilt. Her physical decline since then also impacted the frequency with which she could visit the LTC home. While gender, age, and ethnicity did not significantly affect how caregivers reflected on their burden at the baseline, female caregivers and those over 60 appeared particularly receptive to technological support. This contributes new insights by showing that communication technology, like telepresence robots, plays a role in reducing caregiver burden alongside previously identified determinants like residents’ neuropsychiatric symptoms and caregivers’ competence. Our qualitative findings further support this conclusion, highlighting that relief from guilt and stress, reassurance from LTC homes, and mediation in residents’ care planning—with the support and availability of telepresence robots as a communication technology—all contribute to alleviating the caregiver burden.

A gradual improvement in residents’ quality of life and reduction in loneliness was observed over time in our study, with the most significant changes within the first month. Due to the limited sample size, post hoc comparisons did not reveal statistically significant differences between time points. Another possible explanation is that the stringency of the multiple-comparisons correction renders the pairwise post hoc tests more conservative, resulting in *p*-values slightly above the 0.05 threshold. One research study reported loneliness as closely associated with the quality of life in older adults [40], which aligned with our findings. Although one recent scoping review reported few studies focusing on the impact of assistive technology on the loneliness of residents with dementia [41], one study in this review showed promising results of assistive technologies—not telepresence robots—which could reduce loneliness in people with dementia in LTC [42]. Telepresence robots were reported in one review of their potential to enhance social connection, which contributes to a reduction in loneliness [43]. Our qualitative interviews also echoed this potential through the experiences of using the telepresence robot shared by residents and family caregivers. For example, telepresence robots brought these individuals a sense of closeness, connected residents with their children, grandchildren, and even great-grandchildren, and allowed the sharing of instant life story updates between family caregivers and residents. Besides loneliness, the changes in quality of life were possibly influenced by additional factors such as resident’s baseline health conditions, social engagement, and individual preferences for virtual communication [44,45]. The qualitative data also reported and acknowledged the limitations of the telepresence robots’ impact on residents’ physical health and energy levels despite the other benefits they brought to residents’ and caregivers’ lives. Interestingly, male and white residents reported higher quality of life scores than their female and non-white counterparts, which may reveal cultural or gender-related variations in how residents perceive and engage with telepresence technology, requiring further investigation into individualized interventions tailored to different demographic groups.

The integration of quantitative and qualitative findings in this study offers a more comprehensive understanding of how telepresence robots influence both measurable outcomes and lived experiences. Quantitatively, reductions in caregiver burden and improvements in resident’ quality of life and loneliness were statistically significant or trending toward significance, particularly within the first month of intervention. These shifts were further contextualized and enriched by qualitative data, which illustrated how caregivers experienced immediate emotional relief, reduced guilt, and enhanced connection-mechanisms that help explain the numerical changes observed. Similarly, residents’ improved quality of life and reduced loneliness were reflected in their narratives about feeling more social connected and emotionally supported. These converging findings underscore that the robots’ impact was not merely technological but relational, affecting emotional well-being, family dynamics, and care continuity. By triangulating quantitative measures with in-depth qualitative narratives, the mixed study design effectively addresses the research question.

In terms of practice implications, this study suggests that telepresence robots can offer a valuable tool for supporting family caregivers, particularly older caregivers, who reported the highest initial burden. The sustained decrease in burden over time indicates that continuous use of telepresence robots may offer long-term benefits. Implementing these robots could assist caregivers who face emotional stress and logistical challenges. Given the positive outcomes for both residents and caregivers, policymakers and LTC administrators can consider integrating such technologies to enhance the well-being of residents and their families. This study also offers insights into the target populations that may best benefit from the use of telepresence robots. With limited resources, LTC administrators can explore how these technologies can be best used in their organizations, such as shared use of the robots or for residents who have older adult spouses or children. Furthermore, while the robots facilitate improved social connections, they should be used as a supportive, complementary communication tool rather than replacing the need for in-person visits.

In terms of research implications, further studies should examine the long-term sustainability of the effects observed in this study regarding the reduction in caregiver burden, residents’ level of loneliness, and the improvement in residents’ quality of life. Given the potential differences observed in subgroups, future research should include larger sample sizes and more rigorous study designs to better identify and target the most likely beneficiaries. Also, future studies can investigate the cost-effectiveness of implementing such robots and the staff’s perspectives (those from LTC homes and from different social and cultural contexts). Additionally, since the quality of life may be influenced by multiple factors, such as ethnocultural backgrounds and gender, further exploration of the relationship between telepresence robots and these determinants is warranted to understand their broader impact in detail. Given the significant impact of telepresence robots on the older adult group, researchers could explore the potential challenges of using these technologies (e.g., the technical challenges) and strategies to best support older caregivers in using these robots. Researchers can also further explore the support of telepresence robots for intergenerational bonding between residents and their grandchildren, as well as ethical considerations from residents, family caregivers, and staff with different levels of technological literacy in areas including data privacy and residents’ autonomy.

### Strengths and Limitations

This study utilized a mixed-methods approach, which combines both qualitative and quantitative data to provide a more comprehensive understanding of the research problem. Our research team is composed of a diverse group of multidisciplinary researchers and trainees with expertise in nursing, medicine, occupational therapy, biomedical engineering, and social work, contributing to a more comprehensive investigation. Additionally, the team includes patient partners with dementia and family caregivers, who contributed their lived experience to this study. This engagement helps to ensure that our interventions and findings are relevant, practical, and sensitive to the needs of the participants.

Our study has the following limitations. This pre–post design relied on each participant serving as their own control, which can increase statistical power with smaller sample sizes. However, the absence of a control group means that it is difficult to definitively attribute changes in outcomes solely to the intervention, as other confounding factors may have contributed to the observed improvement. For instance, reductions in caregiver burden and resident loneliness could partly reflect increased attention from the research team, growing familiarity with the technology, or natural fluctuations in caregiver–resident dynamics over time. For the quantitative study part, our sample size was constrained by the number of robots available. We have 20 robots in total. Further, our study was conducted in five different LTC homes based in urban areas of West Canada. The unique context may not be transferrable to LTC homes in rural areas or in other countries. We acknowledge that external validity remains limited, particularly given the sample size, the specific context of participating LTC homes, and the demographic characteristics of our sample. Another limitation is the lack of variations in staff engagement and support and detailed tracking of telepresence robot use frequency across sites (only qualitative data provided), which might have influenced the success of robot implementation and residents’ experience; future research should include usage metrics to better contextualize the impact. Lastly, since finalizing this paper, our team has found out that the company that developed the telepresence robots used in our project has dissolved. Nevertheless, the findings are significant and important enough to highlight the positive outcomes benefiting residents in LTC. The findings can provide insights to support efforts at adopting or developing other similar products to be used in the long term and facilitate the socialization of residents, their families, and friends in LTC homes. Future research should consider randomized controlled trials with larger and more diverse samples across rural and international settings to strengthen generalizability. Additionally, comparative studies exploring different models of telepresence technology and their implementation under varying conditions of staff support and funding mechanisms will be essential to identifying sustainable solutions.

## 5. Conclusions

This two-month telepresence robot intervention demonstrated the potential to reduce the family caregiver burden, enhance quality of life, and lessen loneliness in residents living with dementia in LTC homes. From the residents’ perspectives, the telepresence robot fosters a sense of connection, nurtures closeness, and provides meaningful engagement. From the families’ perspective, telepresence robots enable continuity in care, complement in-person visits, reduce stress and guilt, and offer comfort through reassurance of residents’ well-being, with older caregivers as the greatest beneficiaries. This study offers insights for policymakers and LTC administrators in planning and considering the adoption of similar technologies. Future research could explore the long-term impacts on the well-being of residents and their families, the impact of different factors on residents’ and family care partners’ perceptions of telepresence robots, how to better support older adult caregivers in using telepresence robots, and the robots’ potential in supporting intergenerational connections.

## Figures and Tables

**Figure 1 ijerph-22-00713-f001:**
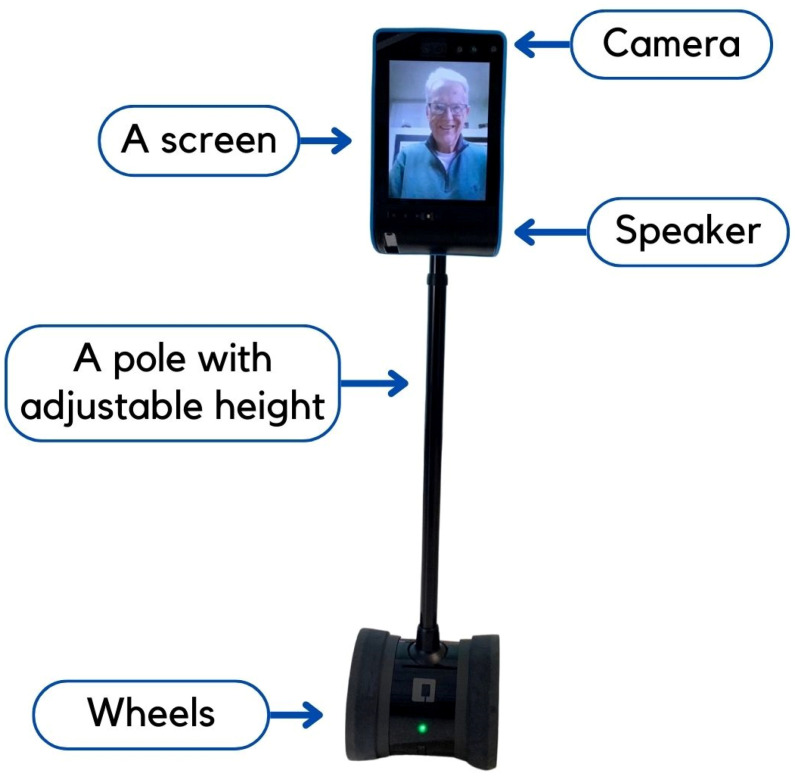
A telepresence robot used in our study.

**Figure 2 ijerph-22-00713-f002:**
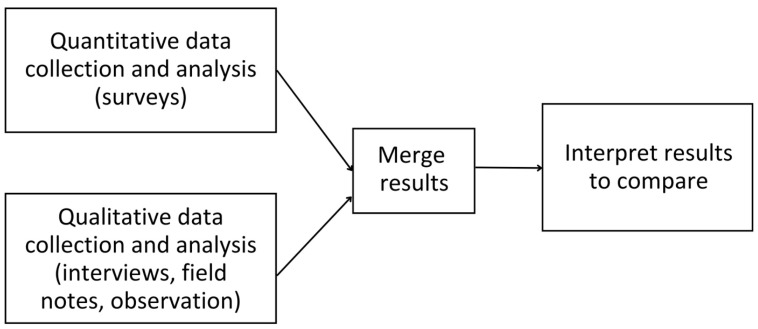
Convergent parallel mixed-methods approach used in this study.

**Figure 3 ijerph-22-00713-f003:**
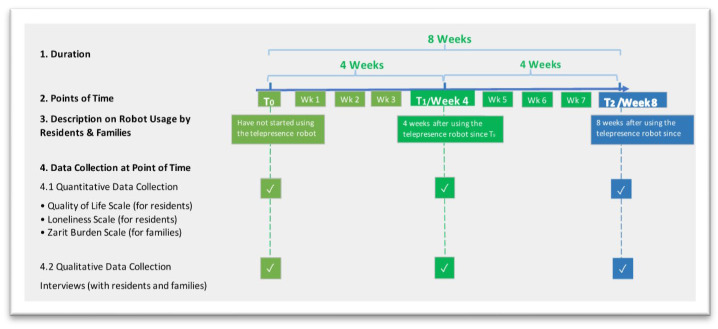
Timeline for data collection for an individual participant.

**Figure 4 ijerph-22-00713-f004:**
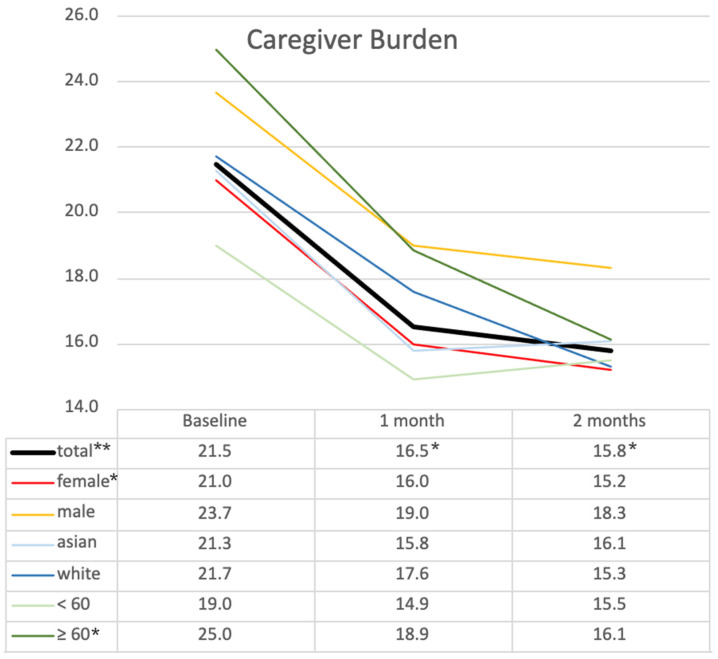
Changes in caregiver burden across demographic subgroups over time. This figure illustrates mean caregiver burden scores at baseline (T_0_), 1 month (T_1_), and 2 months (T_2_) across subgroups based on gender, ethnicity, and age group. * *p* < 0.05, ** *p* < 0.01 (statistical significance determined by ANOVA with post hoc test).

**Figure 5 ijerph-22-00713-f005:**
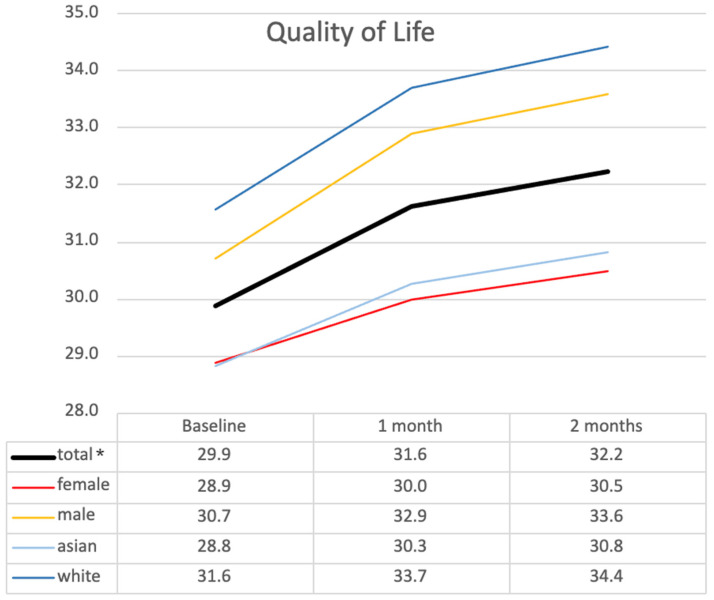
Changes in quality of life across demographic subgroups over time. This figure illustrates mean quality of life scores at baseline (T_0_), 1 month (T_1_), and 2 months (T_2_) across subgroups based on gender and ethnicity. * *p* < 0.05 (statistical significance determined by ANOVA).

**Figure 6 ijerph-22-00713-f006:**
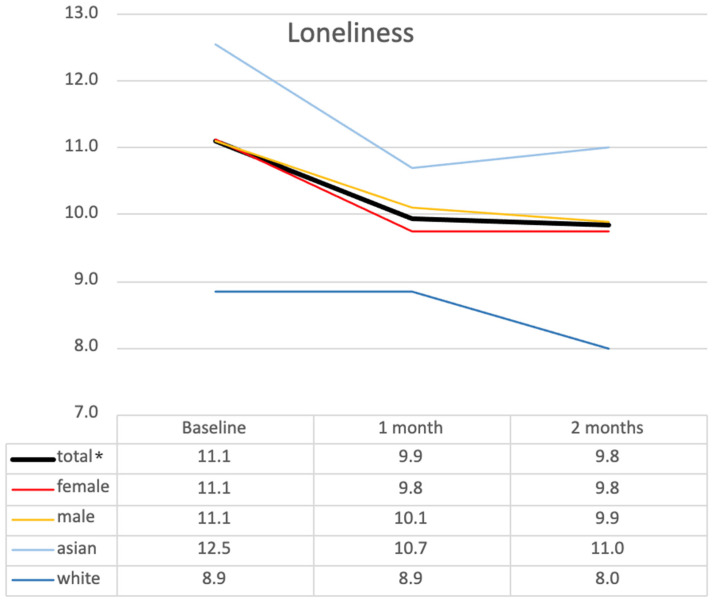
Changes in loneliness across demographic subgroups over time. This figure illustrates mean loneliness scores at baseline (T_0_), 1 month (T_1_), and 2 months (T_2_) across subgroups based on gender and ethnicity. * *p* < 0.05 (statistical significance determined by ANOVA).

**Table 1 ijerph-22-00713-t001:** Detailed information on partnered care sites: types of rooms, staff–resident ratio, and funding.

Pseudonym of LTC Site	Types of Rooms Available	Staff–Resident Ratio	Funding
Jasmine Residence	Individual and Shared Rooms	Day: 1 staff member to 6 residents	Publicly funded
		Evening: 1 staff member to 20 residents	
Rose Residence	Individual and Shared Rooms	Day: 1 staff member to 6 residents	Publicly funded
		Evening: 1 staff member to 8 residents	
		Night: 1 staff member to 25 residents	
Zinnia Residence	Individual Room	Day: 3 nurses and 4 care aides (7 staff members in total to 19 residents)	Publicly funded
		Evening: 3 nurses and 3 care aides (6 staff members in total to 19 residents)	
		Night: 2 nurses and 2 care aides (4 staff members to 19 residents)	
Tulip Residence	Individual Room	Day: 1 nurse and 2 to 3 care aides for 16 to 17 residents	Non-profit
		Evening: 1 nurse and 2 to 3 care aides for 16 to 17 residents	
		Night: 1 staff member (nurse or care aid) per care neighbourhood (16–27 residents)	
Sunflower Residence	Individual Room	LTC: 1 resident care partner for 5–6 residents	Privately funded, for-profit

**Table 2 ijerph-22-00713-t002:** Examples of coding.

Quotes	Codes	Themes
The robot helps with relieving the pressure of driving and commuting. I have two young kids, and it’s hard having the time to drive to him [the resident] with two hours of travelling back and forth. It had been a hard time making time for myself and my kids… and it’s hard to see my dad. The robot visits help save us time for commuting.	Saves travel time Enabling self-care for family caregivers	Reduces Stress and Guilt
She [My granddaughter] will ask me where I feel pain. I feel happy to talk to her through the robot.	Engages residents in the conversation Brings happiness	Provides Meaningful Engagement

**Table 3 ijerph-22-00713-t003:** Demographic characteristics of participants (*n* = 35).

Residents		Families	
**Gender**			
Male	10 (55.6%)	Male	3 (17.6%)
Female	8 (44.4%)	Female	14 (82.3%)
**Age group**			
61–70 years	1 (5.6%)	21–40 years	2 (11.8%)
71–80 years	5 (27.8%)	31–50 years	4 (23.5%)
81- years	12 (66.7%)	51–60 years	4 (23.5%)
		61–70 years	5 (29.4%)
		71–80 years	2 (11.8%)
**Ethnicity**			
White	7 (38.9%)	White	7 (41.2%)
Asian	11 (61.1%)	Asian	10 (58.8%)
		**Relationships**	
		Daughter	11 (64.7%)
		Son	3 (17.6%)
		Spouse	1 (5.9%)
		Granddaughter	1 (5.9%)
		Friend	1 (5.9%)
Total	18	Total	17

**Table 4 ijerph-22-00713-t004:** Changes in caregiver burden, quality of life, and loneliness over time.

	Time Points	Mean ± SD	ANOVA F(df), *p*-Value	Post Hoc Comparison (*p*-Value)
Caregiver Burden	T_0_	21.5 ± 11.09	F(2) = 7.464, *p* = 0.008 **	T_0_ vs. T_1_ (*p* = 0.011) *
	T_1_	16.5 ± 7.92		T_0_ vs. T_2_ (*p* = 0.043) *
	T_2_	15.8 ± 7.68		T_1_ vs. T_2_ (*p* = 1.000)
Quality of Life	T_0_	29.9 ± 6.93	F(2) = 4.024, *p* = 0.027 *	T_0_ vs. T_1_ (*p* = 0.109)
	T_1_	31.6 ± 6.55		T_0_ vs. T_2_ (*p* = 0.124)
	T_2_	32.2 ± 7.62		T_1_ vs. T_2_ (*p* = 1.000)
Loneliness	T_0_	11.1 ± 3.43	F(2) = 4.084, *p* = 0.026 *	T_0_ vs. T_1_ (*p* = 0.120)
	T_1_	9.9 ± 2.99		T_0_ vs. T_2_ (*p* = 0.085)
	T_2_	9.8 ± 2.73		T_1_ vs. T_2_ (*p* = 1.000)

Note: ANOVA *p*-values indicate overall differences across time points. Post hoc comparisons were performed using Bonferroni correction. T_0_ = Baseline, T_1_ = 1-Month Follow-Up; T_2_ = 2-Month Follow-Up. * *p* < 0.05, ** *p* < 0.01.

**Table 5 ijerph-22-00713-t005:** Summary of themes.

Impact on Residents	Impact on Family Caregivers
Promotes connections and nurtures closeness Provides meaningful engagement Acknowledges robots’ limitations	Reduces stress and guilt Enables continuity in care and complements in-person visits Offers reassurance on residents’ quality of life

## Data Availability

The original contributions presented in this study are included in the article/Appendix A. Further inquiries can be directed to the corresponding author.

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
