# Peer review of "The Impact of Telepresence Robots on Family Caregivers and Residents in Long-Term Care"

_ijerph, 2025, doi:10.3390/ijerph22050713_

Round 1
Reviewer 1 Report
Comments and Suggestions for Authors
The study is limited to urban LTC settings in Canada.
Although each participant serves as their own control in this pre-post design, the absence of a separate control group makes it challenging to definitively attribute the observed outcomes solely to telepresence robots. This limitation should be clearly discussed as a potential confounding factor.
While qualitative results acknowledged the robots’ limitations, further details on specific technical or practical issues faced by residents or staff (e.g., operational complexity, privacy concerns) would enrich the discussion and inform practical implementation.
Clarify why post hoc analyses did not show significant differences despite significant overall ANOVA results; discuss potential reasons clearly (e.g., limited statistical power due to small sample sizes).
Enhance the integration of quantitative and qualitative findings explicitly. Although generally well done, explicitly illustrating how qualitative insights explain or expand on the quantitative findings could strengthen the discussion.
Include explicit statements on the future research directions directly connected to observed limitations (e.g., investigation of different robot models or sustainable funding mechanisms).
Include effect sizes explicitly alongside p-values to strengthen the interpretation of the clinical significance of findings.
Section 2.2.4 – Timeline - "For participants who participated in our study in the early stages of the larger project… their answers to these scales were retrospective."
Ambiguity - Table 4 and Section 3.1 (Quantitative Results)ANOVA is reported as significant for Quality of Life (p = 0.027), but post hoc comparisons (T0 vs. T1, p = 0.109) are not significant.
Incomplete caption - Figures 2–4
Comments on the Quality of English Language
Sentences such as “He can get a feel of what is going on in my life” can be rephrased in a more academic tone.
inconsistent terminology - "Residents in care homes" vs. "Residents in LTC homes"
“residents or patients” → consider choosing one term consistently
“yea” (Page 12) → should be formalized to “yes”
Spacing and alignment issues in Table 1
Author Response
|
Comments 1: The study is limited to urban LTC settings in Canada. Response 1: Our study was conducted in urban LTC settings. We have included this limitation in the strengths and limitations session (Page 18). Comments 2: Although each participant serves as their own control in this pre-post design, the absence of a separate control group makes it challenging to definitively attribute the observed outcomes solely to telepresence robots. This limitation should be clearly discussed as a potential confounding factor. Response 2: Thanks for your comment. We agree the limitation of pre-post design and add it in the discussion section (Page 18). Comments 3: While qualitative results acknowledged the robots’ limitations, further details on specific technical or practical issues faced by residents or staff (e.g., operational complexity, privacy concerns) would enrich the discussion and inform practical implementation. Response3: Thank you. We have added an example of technical difficulty faced by a family member and a resident under 3.2.1.3 (Page 14), and added a sentence related implications to future research under discussion (Page 18). |
|
Comments 4: Clarify why post hoc analyses did not show significant differences despite significant overall ANOVA results; discuss potential reasons clearly (e.g., limited statistical power due to small sample sizes). Response 4: Thanks for your comment. We add relevant explanation in the discussion (Page 16). Comments 5: Enhance the integration of quantitative and qualitative findings explicitly. Although generally well done, explicitly illustrating how qualitative insights explain or expand on the quantitative findings could strengthen the discussion. Response 5: Thanks for your comment. We add one more paragraph (Page 17) to address the strength of mixed study on this research question. Comments 6: Include explicit statements on the future research directions directly connected to observed limitations (e.g., investigation of different robot models or sustainable funding mechanisms). Response 6: Thanks for your comments. We add future research directions directly connected to observed limitations. (Page 18). Comments 7: Include effect sizes explicitly alongside p-values to strengthen the interpretation of the clinical significance of findings. Response 7: Thanks for your comments. We add effect sizes in the results in Page 10 -12. Comments 8: Section 2.2.4 – Timeline - "For participants who participated in our study in the early stages of the larger project… their answers to these scales were retrospective." Response 8: Thanks for your comments. We delete this sentence to avoid misunderstanding by readers. Comments 9: Ambiguity - Table 4 and Section 3.1 (Quantitative Results)ANOVA is reported as significant for Quality of Life (p = 0.027), but post hoc comparisons (T0 vs. T1, p = 0.109) are not significant. Response 9: Thanks for your comments. We explain this ambiguity in the discussion as possible limited sample size and more stringent criteria for post hoc comparisons. Comments 10: Incomplete caption - Figures 2–4 Response 10: Thanks for your comments. We add captions for three figures accordingly. |
|
Response to Comments on the Quality of English Language |
|
Point 1: Sentences such as “He can get a feel of what is going on in my life” can be rephrased in a more academic tone. Response 1: Thanks for your comments. We have revised the narratives into a more academic tone while preserving the original wording of direct participants quotes to retain the authenticity. Point 2: inconsistent terminology - "Residents in care homes" vs. "Residents in LTC homes" Response 2: Thank you. We have updated all terms to LTC homes. Point 3: “residents or patients” → consider choosing one term consistently Response 3: Thank you. We have changed all patients to residents. Point 4: “yea” (Page 12) → should be formalized to “yes” Response 4: Thank you. We have updated. Point 5: Spacing and alignment issues in Table 1 Response 5: Thank you. Table 1 updated. |

Reviewer 2 Report
Comments and Suggestions for Authors
This paper presents a study about the impact of telepresence robots on Familiy caregivers and the residents in long-term care. Experiments were conducted by placing telepresence robots in the residents’ rooms and allowing virtual visits at mutual convenience. Quantitative and qualitative data were collected. The results demonstrated telepresence robots can enhance the well-beings of both residents and caregivers. The paper has a good organization, and the method is well introduced. There are also some improvements that should be made.
- The paper lacks introduction about the details of the telepresence robots.
- It is not convincing that the telepresence robots used in this study has a big difference from video phone calls.
- A comparison about the impacts between telepresence robots and regular phone/video calls could be more convincing to show effectiveness about the telepresence robots.
Author Response
Comments 1: The paper lacks introduction about the details of the telepresence robots.
Response 1: Thank you. We have added details of the telepresence robots and a figure in the introduction (Page 2).
Comments 2: It is not convincing that the telepresence robots used in this study has a big difference from video phone calls.
Response 2: Thank you. We have added a comparison in the introduction (Page 2).
Comments 3: A comparison about the impacts between telepresence robots and regular phone/video calls could be more convincing to show effectiveness about the telepresence robots.
Response 3: Thank you. We have added a comparison in the introduction (Page 2).

Reviewer 3 Report
Comments and Suggestions for Authors
The research is quite significant. It measures the impact of telepresence robots on the quality of life of caregivers and residents in long-term care homes. The methodology appears to be sound.
I would like to see more information about the robots mentioned in the study. What does the telepresent robot in the study look like? And how does it interact with both the caregivers and the residents? This information would add to the ease with which the study is understood, though it is not absolutely necessary.
Thus, I suggest that the paper be accepted as is.
Author Response
Comments 1: I would like to see more information about the robots mentioned in the study. What does the telepresent robot in the study look like? And how does it interact with both the caregivers and the residents? This information would add to the ease with which the study is understood, though it is not absolutely necessary.
Response 1: Thank you. We have added more descriptions in the introduction (Page 2).
Reviewer 4 Report
Comments and Suggestions for Authors
The study addresses a timely and socially relevant topic, especially in the post-pandemic context, by exploring the impact of telepresence robots on emotional and social well-being in long-term care (LTC). Integration of mixed-methods research to assess the dual impact on caregivers and residents adds a novel contribution to literature. However, to improve the quality of manuscript, authors must make the following recommendations:
Originality:
1. While the application of telepresence robots in healthcare is not entirely new, this study contributes by empirically validating their impact using a mixed-method approach in the LTC setting. Still, the technological novelty of the robot itself is minimal.
2. The lack of a control group limits the originality from a design standpoint.
Relationship to Literature
3. The review could benefit from a deeper exploration of recent telehealth and AI-driven assistive technologies post-2022.
4. Some gaps in comparative discussion with other telecommunication tools (e.g., tablets, voice assistants) remain unexplored.
Methodology
5. The study lacks a control or comparison group, which limits causal inference.
6. The sample size (n=35) is small, and though the authors justify it based on effect sizes and power analysis, the external validity is limited.
7. Some details of LTC site differences (e.g., staff support, robot use frequency) are not adequately controlled or discussed.
Results
8. Post hoc comparisons show limited statistical significance for some key measures (e.g., loneliness and quality of life).
9. No triangulation of qualitative themes across stakeholder types (residents vs. caregivers vs. staff) is explored.
10. Some statistical interpretations could be better contextualized for practical significance.
Implications for Research, Practice, and/or Society
11. The discussion could further elaborate on cost-effectiveness, sustainability, and staff acceptance of such interventions.
12. Ethical concerns such as data privacy, resident autonomy, and technological literacy are acknowledged only minimally.
Quality of Communication
13. There are minor instances of redundancy (e.g., repeated mention of the same experiences) and a few grammatical issues that require minor editing.
14. Some sections (particularly in the qualitative results) could benefit from more concise synthesis.
Comments on the Quality of English Language
Quality of Communication
13. There are minor instances of redundancy (e.g., repeated mention of the same experiences) and a few grammatical issues that require minor editing.
14. Some sections (particularly in the qualitative results) could benefit from more concise synthesis.
Author Response
Originality:
Comments 1: While the application of telepresence robots in healthcare is not entirely new, this study contributes by empirically validating their impact using a mixed-method approach in the LTC setting. Still, the technological novelty of the robot itself is minimal.
Response 1: Thanks for your comments. We acknowledge that our study’s contribution lies not in the novelty of the technology itself, but in the application of mixed-method design to empirically assess its impact in the specific environment of LTC. Rather than emphasizing technological innovation, our study highlights contextual integration and human-centered outcomes, offering practical insights for implementation in similar care environment (Page 16).
Comments 2: The lack of a control group limits the originality from a design standpoint.
Response 2: Thanks for your comments. We add limitations and possible confounding factors of pre-post design in the discussion section (Page 17).
Relationship to Literature
Comments 3: The review could benefit from a deeper exploration of recent telehealth and AI-driven assistive technologies post-2022.
Response 3: Thanks for your comments. We explore more on recent telehealth and AI-driven assistive technologies after 2022 in the introduction section (Page 3).
Comments 4: Some gaps in comparative discussion with other telecommunication tools (e.g., tablets, voice assistants) remain unexplored.
Response 4: Thank you. We have added a comparison of telepresence robots with regular phone and video calls in the introduction (Page 2).
Methodology
Comments 5: The study lacks a control or comparison group, which limits causal inference.
Response 5: Thanks for your comments. We add limitations and possible confounding factors of pre-post design in the discussion section (Page 17).
Comments 6: The sample size (n=35) is small, and though the authors justify it based on effect sizes and power analysis, the external validity is limited.
Response 6: Thanks for your comments. We address the limited external validity in the discussion section (Page 17).
Comments 7: Some details of LTC site differences (e.g., staff support, robot use frequency) are not adequately controlled or discussed.
Response 7: Thanks for your comments. We acknowledge that robot use frequency was not accurately recorded, and staff support was only reflected in the qualitative data without well well-controlled in this pre-post research design. We have added this limitation to the discussion section. (Page 17).
Results
Comments 8: Post hoc comparisons show limited statistical significance for some key measures (e.g., loneliness and quality of life).
Response 8: Thanks for your insightful observations. We add a possible explanation in the discussion section (Page 15).
Comments 9: No triangulation of qualitative themes across stakeholder types (residents vs. caregivers vs. staff) is explored.
Response 9: Thank you. In our previous publications, we reported on the perspectives of staff, residents and family caregivers using qualitative themes towards the implementation of the robot. These papers are available as follows.
Staff’s perspectives: https://bmcnurs.biomedcentral.com/articles/10.1186/s12912-024-01983-0
Experiences of residents and family caregivers:
https://journals.sagepub.com/doi/full/10.1177/20552076251319820
https://journals.sagepub.com/doi/full/10.1177/23337214231166208
Our large project did not aim to “verify” a single truth. Instead, what is more important to us is to explore the diverse experiences and perspectives of stakeholders from different groups to gain a more comprehensive understanding of implementing telepresence robots in LTC homes, a care environment of complexity. Understanding multiple realities offers insights into the diverse needs and priorities of residents, family caregivers and staff. These findings can inform researchers, policymakers, clinical educators and clinicians in their future research and practices to be better tailored to the needs of different stakeholders.
Comments 10: Some statistical interpretations could be better contextualized for practical significance.
Response 10: Thanks for your comments. We add effect size to convey the magnitude of observed effects. These values help illustrate the real-world relevance of our findings, especially when paired with the qualitative narratives that describe reductions in stress, guilt, and increased connection.
Implications for Research, Practice, and/or Society
Comments 11: The discussion could further elaborate on cost-effectiveness, sustainability, and staff acceptance of such interventions.
Response 11: Thank you. We added more implications on cost-effectiveness and sustainability (Page 20). Staff’s perspectives have been reported in a separate paper published earlier, which is available at: https://bmcnurs.biomedcentral.com/articles/10.1186/s12912-024-01983-0
Nevertheless, we added staff’s perspectives from different LTC homes as you suggested.
Comments 12: Ethical concerns such as data privacy, resident autonomy, and technological literacy are acknowledged only minimally.
Response 12: We added ethical considerations, such as data privacy and technological literacy, on page 20. Resident autonomy has been reported in a separate paper cited in this paper (doi:10.1177/20552076251319820 ).
Quality of Communication
Comments 13: There are minor instances of redundancy (e.g., repeated mention of the same experiences) and a few grammatical issues that require minor editing.
Response 13: Thanks for your comments. We have corrected the redundancy and grammatical issues through the whole manuscript.
Comments 14: Some sections (particularly in the qualitative results) could benefit from more concise synthesis.
Response 14: Thanks for your comments. We have made the synthesis more concise.
